# Water Chemistry of Arctic Lakes under Airborne Contamination of Watersheds

**Tatyana I. Moiseenko [1],*** , **Natalia A. Gashkina [1]**, **Marina I. Dinu [1]**, **Tatiana A. Kremleva [2]** and **Vitaliy Yu. Khoroshavin [2]**

[1] V.I. Vernadsky Institute of Geochemistry and Analytical Chemistry RAS, Kosygina Street 19, 119991 Moscow, Russia; ngashkina@gmail.com (N.A.G.); marinadinu999@gmail.com (M.I.D.)

[2] Department of Organic and Ecological Chemistry, Tyumen State University, Lenina Street 25, 625003 Tumen, Russia; t.a.kremleva@utmn.ru (T.A.K.); v.y.khoroshavin@utmn.ru (V.Y.K.)

\* Correspondence: moiseenko.ti@gmail.com

**Abstract:** The data on the metal contents and acidification of small lakes caused by airborne contamination of the watershed in three industrial regions of the Arctic—European Russia (Kola region), Western (Yamal-Nenets region) and Eastern Siberia (Norilsk region)—have been presented for the first time. It has been proven that acidification and enrichment by metals of water connect with sulfur dioxide and metals emissions from copper–nickel smelters, contaminating the catchments, with associated gas burning during raw hydrocarbon production. To assess the effects of acid deposition, critical loads and their exceeds were calculated: exceeded by 56% and 12.5%, respectively, in lakes in the Kola region and in the north of Western Siberia; the catchments of the East Siberian region are resistant to acidification. Water enrichment factors (EF) by elements were calculated to show that the waters of the Norilsk and Kola regions are enriched with Ni, Cd, As, Sb and Se as a result of emissions from copper–nickel smelters. The oil and gas industry in the northern regions of Western Siberia lead to the increase in V, Pb and Mo concentrations in the waters. The high values of EF and excess of acidity critical loads for water are explained by the local and transboundary pollution impacts on the catchment of small lakes.

**Keywords:** Arctic lakes; water chemistry; catchment pollution; acidification; metal enrichment

## 1. Introduction

The Arctic regions are particularly vulnerable to anthropogenic factors due to the low level of mass and energy exchanges in cold latitudes, the rapid moving of ecotoxicants in short food chains and the high sensitivity of organisms to adverse effects. Climate warming, and local contamination and transboundary transfer of contaminants from the atmosphere are an environmental threat to these sensitive ecosystems [1,2]. An analysis of publications on surface water in the Arctic regions shows that the main attention of the scientific community in recent years is aimed at studying the impact of climate on water resources and changing their quantity [3–6]. However, with a high quantitative supply of water to the Arctic regions, we may encounter a qualitative depletion of water.

It has been proven that the transboundary transfer of contaminants from Asia, Europe and North America as well as local sources of atmospheric emissions significantly contribute to the contamination of the Arctic regions [7,8]. The migration of sulfur dioxide to the Arctic regions and local sources of emissions from non-ferrous metallurgy smelters has been proven to result in water acidification [9]. Elevated concentrations of hazardous elements such as Hg, Cd and Pb have been found in the snow cover of Greenland and the High Arctic [10]. Close attention has been paid to the biogeochemistry of mercury in the Arctic regions [11].

Most of the articles are devoted to studying the influence of industry and wastewater on the surface waters of the Kola region. In the work [12], using the example of the Kola region, it was shown that Arctic waters develop anthropogenic (induced) processes, such as acidification, toxic pollution and eutrophication of waters, emphasizing the high sensitivity of the geochemistry of waters and ecosystems to the flow of pollutants from catchments. It is also noted that in the Kola North, under the influence of industry, the content of metals increased and concentrations of the main cations of alkali and alkaline earth metals, N compounds, total dissolved solids and heavy metals were found in the lakes, which indicate exposure to anthropogenic impacts [13]. Using the example of natural conditions in the northern low-salinity freshwaters of the Kola Peninsula, it is demonstrated that the labile forms of most metals are the most bioavailable and toxic.

Studying the effect of atmospheric deposition on catchments and lakes requires special methodological approaches. The consequence of the predominance of precipitation over evaporation in the Arctic is the presence of a large number of small oligotrophic ultrafresh lakes, the supply of which is provided by atmospheric deposition to catchments. Weak vegetation development and thin soil cover provide high drainage of precipitation and contaminants, which makes atmospheric precipitation on catchments a determining factor in the formation of the chemical composition of lake waters.

The land of the Arctic zone is reached with numerous small lakes of predominant atmospheric feeding, which makes up over 70–95% of the total water balance of small lakes. Provided that small lakes are no direct sources of contamination, they are good indicators of the airborne impacts on the watershed of acid deposition and metals on a regional and global scale. However, the data on water acidification and metal contaminations of the surface water in the Arctic are limited and date back to the early 2000s. In addition, the contaminants emissions into the atmosphere are rapidly reducing in local contamination zones [8].

The airborne pollution in the Arctic regions of Russia has its own specificity. In the Kola and Norilsk regions, copper–nickel smelters of the Nornikel Smelter are concentrated. Sulfur dioxide and such elements as Ni, Cu, Cd and others are distributed with the emissions of the smelter and deposition on the watershed. Sulfur dioxide emissions entail acid rain occurrence, which causes water acidification and more intensive metals leaching into the lake systems. Climatic features determine the spread of emissions of Norilsk-Talnakh industry at a distance of 500–600 km to the west, to the Gydan Peninsula [9,10].

Oil and gas production in the Yamal-Nenets District of Western Siberia is a regional source of contamination. The sulfur content in oil is relatively low (up to 1 mass %), with almost no mercaptans and disulfides. Nitrogen compounds, such as pyridines, amides, imides and pyrrole-benzene derivatives, have a significantly larger mass in the oil and associated petroleum gases. The mass fraction of NOx can reach up to 9% in associated petroleum gas. The burning of associated petroleum gas leads to the formation of acidic deposition, mainly due to nitrogen oxides [11]. Trace amounts of metals such as V, Ni, Fe, Al, Cu, Sr, Mn, Co, Mo, Cr, As, Mo and others are found in the oil produced [12]. These elements are part of organic and inorganic compounds of oil. Oil production leads to the enrichment of surface waters with metals, including the regions of the Arctic Basin [13].

The purpose of this paper was to determine the water chemistry of Arctic lakes under airborne contamination of watersheds by acid compounds and metals deposition, and to assess the critical levels of acid deposition and anthropogenic increase in the content of elements in the surface water in the present period.

## 2. Materials and Methods

The paper is based on the study results' generalization of the chemical composition in tundra small lakes (ER, Kola region—40 lakes, 2014), Western Siberia (WS, Yamal-Nenets region—52 lakes, 2013–2014) and Eastern Siberia (ES, Norilsk region—12 lakes, 2016).

All the studied lakes are located outside of any direct sources of contamination and are susceptible to airborne contamination. The points of sampling are shown in Figure 1. The studies included

lakes remote from direct sources of pollution, and mainly with an area of 0.4–20 km$^2$. In a short time (September–October), water samples were taken from the lakes to conduct chemical analysis.

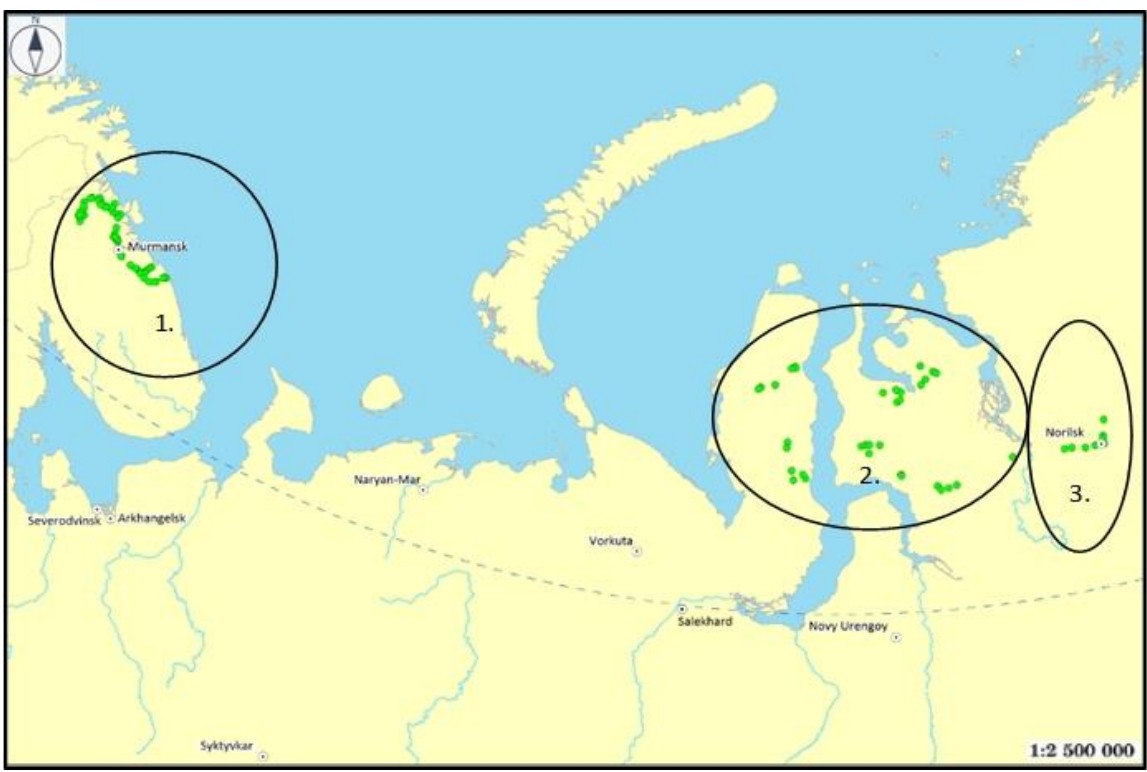

**Figure 1.** Regions of investigation and sampling points: Northern part of Euro-Asian territory of Russia. The dotted line is the border of the Arctic Circle. Points indicate selected lakes. 1—European Russia, 2—Western Siberia, 3—Eastern Siberia.

Water sampling was carried out into plastic bottles, the material of which does not have sorbing properties. The bottles were thoroughly cleaned in the laboratory in advance. When water sampling, the bottles were rinsed twice with lake water, then placed into dark containers, refrigerated (~ +4 °C) and quickly transported to the laboratory. Chemical analyses of water samples were performed according to unified methods in accordance with the recommendations [14]. Water samples from the lakes of the Kola Arctic region were analyzed in the laboratories of the Institute of Industrial Ecology of the North, KSC RAS, the samples from Norilsk at the Vernadsky Institute of Geochemistry and Analytical Chemistry of RAS and in Western Siberia, at the Laboratory of Water Quality of the Tyumen University using unified methods. Analytical studies included determination of pH, electrical conductivity, base cations (Ca$^{2+}$, Mg$^{2+}$, K$^+$, Na$^+$), Si, alkalinity (Alk), SO$_4^{2-}$, Cl$^-$, color (Col), organic matter content (DOC), total nitrogen (TN) and its forms (NO$_3^-$ and NH$_4^+$), total phosphorus TP and PO$_4^{3-}$. The verification of analytical methods and the results of the water chemical composition determining was carried out under constant tight internal laboratory control using a unified system of standard solutions.

Element concentrations in aqueous samples (B, Li, Be, Al, Sc, Ti, V, Cr, Mn, Co, Ni, Cu, Zn, Ga, Ge, As, Se, Br, Rb, Sr, Y, Zr, Nb, Mo, Ru, Rh, Pd, Ag, Cd, In, Sn, Sb, Te, Cs, Ba, La, Ce, Pr, Nd, Sm, Eu, Gd, Tb, Dy, Ho, Er, Tm, Yb, Lu, Hf, Ta, W, Re, Os, Ir, Pt, Au, Hg, Tl, Pb, Bi, Th and U) were determined using the ICP-MS method (X-7 ICP-MS Thermo Electron). The elements, the values of which were predominantly below the threshold of analytical measurements, were not considered during the data interpretation. The verification of analytical methods and the results of determining the chemical composition of water were carried out under constant tight internal laboratory control using a unified system of standard solutions.

Statistical data processing was carried out using k-mean ++ clustering methods, taking into account data enumerating and centering changes.

The averages for each cluster were calculated for certain parameters to assess differences from each other. Very different averages were obtained for most used in the analysis. The F statistics obtained for each measurement were an indicator of how well the corresponding measurement discriminated against the clusters.

Electrical conductivity, pH, color and representatives of elements of various groups (alkaline earth, alkaline, transition metals, lanthanides and actinides) are assessed as the selected evaluation parameters. The software Statistica 12 was used.

## 3. Results

### 3.1. The Main Characteristic of Water Chemistry of Small Lakes

Table 1 shows the chemical composition of the tundra lakes in the three studied regions.

**Table 1.** Median and variability of the general chemical parameters (min–max) in the water of lakes in the Arctic regions: the European part of Russia (ER, Kola region), Western Siberia (WS, Yamal-Nenets region) and Eastern Siberia (ES, Norilsk region).

| Parameter | Tundra/Forest-Tundra Regions | | |
| --- | --- | --- | --- |
| | Kola Region, n = 40 | Yamal-Nenets Region, n = 52 | Eastern Siberia, n = 12 |
| pH | 6.43 (5.52–7.00) | 6.34 (4.81–7.39) | 8.0 (6.7–8.7) |
| Cond., $\mu S\ cm^{-1}$ | 35.3 (22.1–109) | 27.7 (9.1–166) | 190 (50–390) |
| Ca+Mg, $mgL^{-1}$ | 2.18 (0.82–12.8) | 3.84 (0.96–13.6) | 27.0 (6.50–70.7) |
| Na+K, $mg\ L^{-1}$ | 3.82 (2.53–10.3) | 3.06 (1.33–21.0) | 3.19 (1.00–9.50) |
| Alk, $\mu eqL^{-1}$ | 74.0 (6.00–391) | 232 (70–620) | 616 (150–2270) |
| Color, Cr/Co | 22.0 (5.00–88.0) | 19.6 (1.3–160) | 11 (2–30) |
| $P_{общ,}$ $\mu g\ L^{-1}$ | 6.0 (1.0–21) | 45 (4–189) | 68.5 (52–571) |
| $N_{общ,}$ $\mu g\ L^{-1}$ | 153 (61.0–684) | 610 (110–2340) | 20 (14–101) |
| $S_{общ,}$ $mgL^{-1}$ | 1.20 (0.26–2.10) | 0.21 (0.07–2.08) | 6.8 (1.3–88) |
| $C_{орг,}$ $mgL^{-1}$ | 5.22 (2.97–12.5) | 5.45 (1.3–14.6) | - |
| COD Mn, $mgO\ L^{-1}$ | 4.80 (1.86–14.3) | 3.67 (0.73–50.6) | 12.1 (1.63–29.6) |

The catchments of the Kola region lakes are composed of magmatic and crystalline rocks, which have the highest resistance to element leaching. The waters of these lakes are characterized by low salinity and oligotrophic nature, which are caused by mostly atmospheric feeding thereof, the development of geological formations resistant to chemical weathering, a thin soil cover (mostly tundra gley and tundra podzolized brown soils) and poor vegetation. The waters of 94% of local lakes are sodium-chloride due to the influence of marine aerosols from the Barents Sea.

Despite the fact that the lakes remote from the copper–nickel smelters that operate on the Kola Peninsula were included into consideration, the appearance of lakes where sulfates dominate is explained by the region-wide increased level of technogenic sulfates deposited on catchments. Anthropogenically acidified lakes are found here, characterized by low alkalinity and relatively high contents of technogenic sulfates and low water pH (<6).

Based on the content of organic matter and nutrients, we can conclude about the low trophic status of lakes—ultra-oligotrophic lakes are common. In some forest-tundra swampy landscapes, lakes with very high water color (up to 80 Pt) are found. Such lakes are characterized by low pH values due to high concentrations of humic acids.

The catchments of Western Siberia are composed of Quaternary sedimentary rocks of marine and glacial origin. These are frozen, monomineral rocks with a large proportion of silicates and quartz sands.

Therefore, the lakes of the tundra zone in the WS, as in the ER, have low-salinity waters (Table 1), and a low content of dissolved organic matter and nutrients. Atmospheric precipitation and fresh water formed during the seasonal thawing of permafrost are the main feeding source for tundra lakes of WS. The high variability in the salt content value may be explained both by the difference in the lakes' water salinity in the areas of distribution of marine clay sediments and glacial loamy sands, and by the influence of aerosol transfer from the Kara Sea.

The variability in pH value is closely related to the swampiest of catchments and their location relative to the seacoast. $HCO_3^-$ predominates among the salinity ions in the waters of the WS tundra and forest-tundra lakes, and $Na^+$ and $Ca^{2+}$ are contained in approximately equal, very low amounts—2.00 and 3.00 mg $L^{-1}$, respectively. It should be noted that the total content of basic ions decreases when moving towards the forest-tundra zone, where waters become less saline. Elevated nitrogen and lower phosphorus contents are characteristic for the WS waters in the lakes in comparison with the waters of the Kola North.

The catchments of the Norilsk region are composed of coarse rock material of basic composition (basalts), with permafrost-taiga and mountain-tundra soils forming thereon. The lakes of the Norilsk region significantly differ from the lakes of the Kola and Yamal-Nenets regions in terms of the chemical composition of the waters, primarily due to higher salinity and pH values. This is partly due to the location of the studied lakes in the vicinity of the industrial centers of the Norilsk plant. The content of salts and phosphorus is significantly—over 10 times—higher than those in the lakes of other studied territories. Sulfates have the highest influence in the Norilsk areas: reaching 20 mg $L^{-1}$ in lakes. Higher salinity and phosphorus contents in the waters of the local lakes occur with the proximity of the location of the lakes to the Norilsk smelter.

To sum it up, it is obvious that despite the general belonging to the Arctic zones, the lakes, along with the general characteristics, have their own peculiarity, influenced by the geological structure and anthropogenic impacts. The lakes of the Kola region and Western Siberia may be sensitive to acidification, in terms of alkalinity and pH variability.

## 3.2. The Content of Trace Elements in the Waters

The content of trace elements in the surface waters of the Arctic zones is presented in Table 2. The Hg content in water was below the analytical detection threshold in our studies. Elements show specific features of their distribution despite significant variability in the concentrations of the majority of them in all the Arctic regions from our research. The highest concentrations of Pb were found in the waters of the lakes around the Norilsk plant, which is explained not only by the transboundary transfer but also, to a greater extent, by the proximity of the lakes to industrial centers. These values are very low both in the Arctic lakes of ER and WS. This watershed is far away from many highways.

Interesting patterns of Cd distribution are observed: in the waters of the Kola region, its concentrations are higher, which may be due to Cd leaching by acid deposition. Higher concentrations of other trace elements are also observed here: the behavior of Se and Cs is manifested in the same way as that of Cd. Higher contents of the lanthanides and alkaline earth elements are also observed in the lake waters of the Kola region.

The contents of Ni and Cu have high variations in the water of each studied region. At the same time, the Ni content is greater in the waters of the lakes around the Norilsk smelter, as well as in the waters of the Kola region, which is caused by the influence of the Nornickel smelters on the watershed. Elevated Cu content is typical for the waters of the WS North. The Al content is high in the ER northern waters, where the underlying rocks are composed of granite-gneissic formations. It is known that the acid precipitation recorded here leads to more intensive leaching of this element from the catchment rocks.

The Fe content (in median values) is quite similar in all the examined regions of the Arctic Basin, however, its variability in the WS waters is maximal—from 10 µg $L^{-1}$ to over 1 mg $L^{-1}$. Cu, Vo, V, Sb

and As also showed higher contents in WS. We also found high concentrations of Bi in the waters of the northern territories, which in some lakes exceeds the Pb content.

The analysis of the contents of trace elements, including heavy metals in the waters of lakes of airborne contamination, showed that their concentrations are relatively low on average. However, the analysis of the greatest values has shown that their occurrence is associated with anthropogenic factors.

**Table 2.** Median and variability in trace element concentrations (min–max) in the water of lakes in the Arctic regions.

| Elements | Tundra and Forest-Tundra | | |
|---|---|---|---|
| | Kola Region, n = 40 | Yamal-Nenets Region, n = 52 | Eastern Siberia, n = 12 |
| | µg L$^{-1}$ | | |
| **Al** | **54.0 (13.7–180)** | 19.9 (10–310) | 14.4 (<3.5 – 79.9) |
| Fe | 50.0 (4.3–600) | 57 (10–1474) | 65.5 (<4 – 227) |
| Ti | 1.79 (0.48–8.27) | <0.6(<0.6–4.8) | 0.8 (<0.3 – 2.2) |
| Mn | 1.6 (0.2–18.0) | 6.7 (1.1–26.1) | 10.8 (<3.11 – 37.4) |
| Sr | 11 (4–23) | 4.70 (0.70–34.8) | 65.4 (13.3 – 115) |
| Zr | 0.05 (<0.02–0.15) | 0.03 (<0.01–0.42) | 0.1 (<0.02 – 0.12) |
| Rb | 0.63 (0.34–1.73) | 0.37 (0.16–0.947) | 0.5 (0.04 – 5.63) |
| V | 0.35 (<0.02–0.76) | 0.11 (<0.04–1.0) | 0.4 (0.15–0.82) |
| Zn | 0.9 (0.2–4.7) | 4.33 (1.94–20.0) | 4.3 (<0.30–10) |
| Cr | 0.2 (<0.1–0.5) | 0.4 (<0.4–0.5) | 0.2 (<0.05–0.22) |
| Ce | 0.29 (0.12–1.13) | 0.07 (0.02–0.56) | <0.01 (<0.01–0.09) |
| Ni | 0.9 (<0.2–5.5) | 0.93 (<0.2–2.78) | 4.3 (0.73–15.1) |
| Cu | 0.7 (0.2–2.9) | 2.58 (1.23–9.02) | 6.3 (2.5–20.5) |
| Li | 0.18 (0.11–0.65) | 0.84 (0.32–4.14) | 0.4 (0.14–1.64) |
| La | 0.24 (0.07–0.62) | 0.04 (0.02–0.17) | <0.007 (<0.007–0.04) |
| Co | <0.2 (<0.2–0.3) | 0.06 (<0.04–0.19) | 0.1 (0.03–1.02) |
| Pb | <0.1 (<0.1–0.6) | 0.30 (0.11–3.39) | 1.4 (<0.38–1.54) |
| Sc | 0.4 (<0.1–0.8) | <0.04 | - |
| Sn | 0.09 (0.04–0.32) | <0.01 (<0.01–0.13) | <0.01 |
| U | 0.03 (0.01-0.30) | 0.04 (0.02–0.06) | <0.01 (<0.01–0.13) |
| As | 0.1 (<0.1–0.3) | 0.43 (0.10–1.56) | 0.1 (<0.002–0.19) |
| Mo | 0.14 (0.04–0.32) | 0.93 (0.25–6.26) | 0.1 (<0.02–0.27) |
| Sb | 0.04 (0.01–0.13) | 0.16 (0.12–0.27) | <0.01 (<0.01–0.13) |
| Cd | 0.09 (<0.05–0.21) | 0.01 (<0.01–0.08) | <0.01 (<0.01–0.04) |
| Bi | 0.01 (<0.01–0.03) | 0.45 (0.13–3.86) | <0.01 |

## 4. Discussion

### 4.1. Clustering of Waters of the Studied Regions Based on Their Chemical Composition

To classify the studied waters according to their chemical composition, k-clustering was applied. The method involves the identification of reliable relationships within a given cluster (the territorial location of waters in our case). Significant reliability of the results was achieved using data sequence enumerating. This allowed us to identify the elements which are the most obvious markers of similarity and differences in the chemical composition of the studied waters.

Parameters pH, Ca+Mg, Cu, La, Mo and Ce contribute the most to the clustering (most reliable F-test value). The most statistically favorable cluster divisions (with the difference p within the group being less than $10^{-5}$) were visualized (Figure 2).

The results obtained using statistical processing indicate a greater difference between the ES lakes and the waters of the other Arctic regions (WS and ER), which is explained by their proximity to industrial centers.

The analysis also demonstrated sufficient statistical convergence of the ER and WS waters in a number of parameters, such as transition elements (Ni, Cu) content, Fe, alkaline earth metals and lanthanides. Notably, some components are most conducive to increasing cluster similarity within each selected cluster—the content of Al, Cr and Sc.

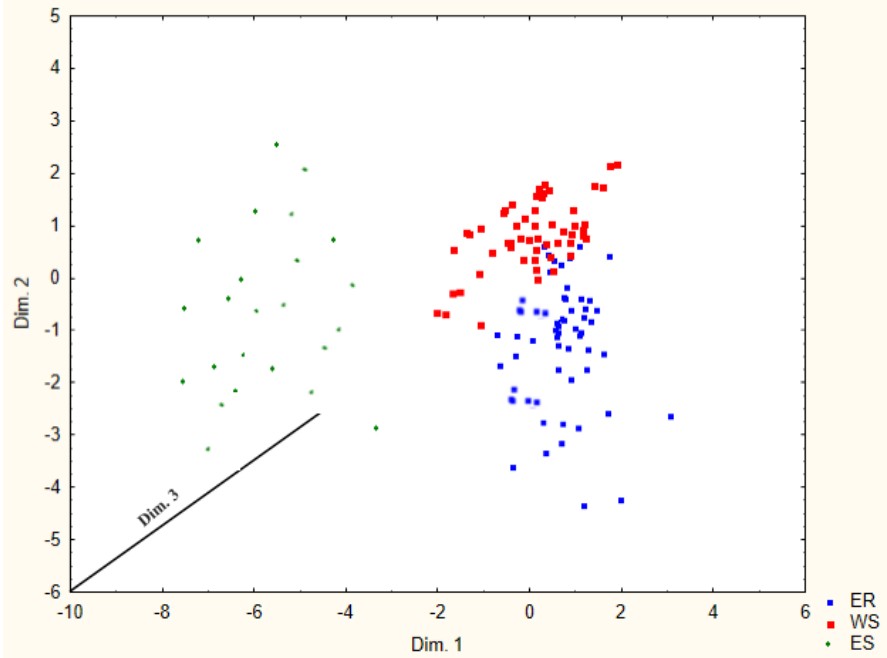

**Figure 2.** K-means clustering for the most representative parameters: pH, conductivity, color, elements of alkali and alkaline earth metals, from the group of heavy metals and scattered elements.

### 4.2. Vulnerability and Water Acidification Assessment

The local sources of acid deposition of the Arctic region are the metallurgical enterprises of the Norilsk copper–nickel smelters (ER and ES) and also the combustion of associated gas during hydrocarbon production in the north of WS, as well as the transfer of acid-forming agents [4–6,15–21]. Using the Kola subarctic zone as an example, it was first proved that water acidification in the Russian Arctic is manifested as the following: (a) a decrease in the pH value of waters in small lakes; (b) the development of "pH shock" during the flood period of streams; and (c) a decrease in the buffer capacity of catchments [17–22]. Among the 460 lakes examined in the Kola North in 1995 within the Northern European Lake Survey (Finland, Norway, Sweden, Denmark, Russian Kola, Russian Karelia, Scotland and Wales) project, the anthropogenic acidified lakes, with pH <6, amounted to 26%, of which 11% had pH values of 5 [23]. The acidification mechanism is described in the above works and is associated with the replacement of bicarbonates in the water with stronger technogenic acids (sulfates and nitrates), as a result of which the buffer capacity of water decreases.

High vulnerability to acid deposition on the watershed is typical not only for the Kola region, but also for the tundra Arctic and subarctic thin soils of Western and Eastern Siberia, lying farther to the east [24]. Acid deposition on the watershed in the northwest of Russia (Kola region) is estimated at 10–12 meq $(m^2)^{-1}$ year (EMEP / MSC − W 2000), and in Eastern and Western Siberia, at 10–15 meq $(m^2)^{-1}$ year [25,26]. In the past decades, sulfur dioxide ($SO_2$) emissions have declined in Europe and North America [19], as well as in Russia [20]. However, the global impact of $SO_2$ and $NO_x$ emissions from developing countries such as China remains [27–38]. Therefore, it is important to assess the current state of acidification of waters in new areas.

Generalization of our recent data for three Arctic regions has shown that lakes with low pH values and low water color are found in such Arctic regions as ER and WS. In the ER tundra zone, the lakes

with pH < 6 and color of 10°Pt − Co scale were 4.4%; in WS such lakes, 8.2% of the examined lakes. In ES, no such lakes were found.

Acid-neutralizing capacity (ANC) is the internationally accepted criterion for assessing water acidification [28,30]. For the waters of the Kola region, an approach developed for the Scandinavian waters caused by the similarity of the geological structure of the Baltic Shield is possible. ANC is calculated as

$$\text{ANC}_1 = \text{Ca}^{2+}{*}+\text{Mg}^{2+}{*}+\text{Na}^+{*}+\text{K}^+{*}-\text{SO}_4{}^{2-}{*}-\text{NO}_3{}^-, \tag{1}$$

The content of elements is presented as equivalents after correction for marine Cl according to the method described in the papers (designated ∗).

However, the assumption that Cl in freshwaters is mainly of marine origin is not true for the WS waters. The paper [12] showed that chlorides in the WS waters can have a geological origin. Therefore, the correction of the main salinity ions for marine aerosols of the WS waters does not adequately characterize the acid-neutralizing capacity of waters. An equation without correction for marine aerosols was used to calculate the ANC for the WS waters:

$$\text{ANC}_1 = \text{Ca}^{2+} + \text{Mg}^{2+} + \text{Na}^+ + \text{K}^+ - \text{SO}_4{}^{2-} - \text{NO}_3{}^- - \text{Cl}^-, \tag{2}$$

The results of the ANC determination for the waters of the three Arctic regions are presented in Figure 3. The waters of the tundra zone of the Kola region and in WS show low values, while the lowest values of this indicator are found in the waters of the Kola region. WS also has lakes with low ANC values. We do not undertake to assess the state of the acidification of the Arctic regions of ES, however, the lakes near the Norilsk plant have extremely high ANC values, which indicates a high buffer capacity of the watershed.

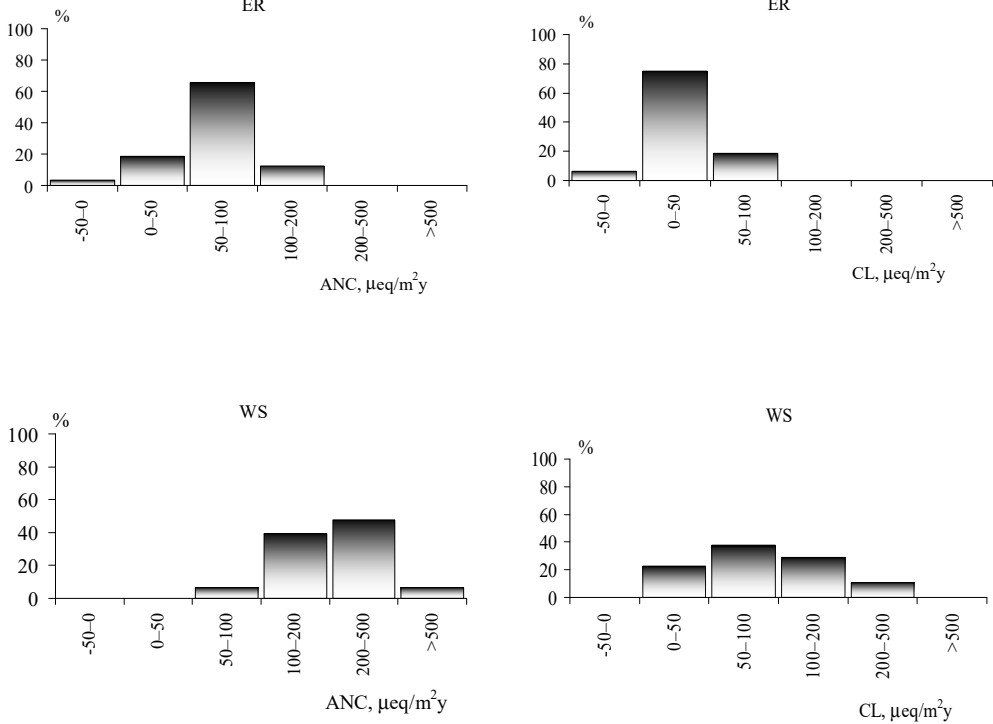

**Figure 3.** *Cont.*

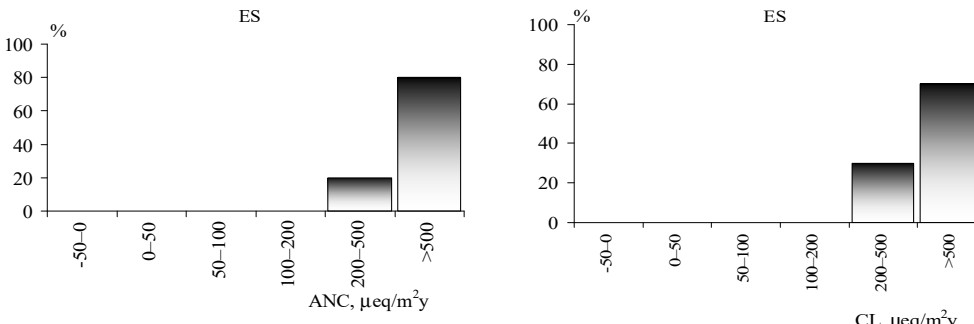

**Figure 3.** The percentage distribution of lakes according to the values of water acid-neutralizing capacity (ANC) and critical loads (CL) in the Arctic regions of the European territory of Russia (ER), Western (WS) and Eastern Siberia (ES).

### 4.3. The Critical Value of Anc for the Studied Regions

Water acidification affects the biological communities of aquatic ecosystems. As the ANC of the waters of the Kola North decreases to 50 $\mu$eqL$^{-1}$ (corresponding to a pH of 5.8–6.2), the proportion of acidobiontic and acidophilic species of diatoms increases, and lake zooplankton become dominated by acidification-resistant species, from the bottom communities: amphipods, gastropod mollusks and nymphs disappear. Amphipod, being a typical inhabitant of the natural oligotrophic waters of the Arctic Basin, is not found in the lakes with water pH < 6.5 [23]. It should be noted that Arctic waters are inhabited by Arctic char, trout and whitefish, which are extremely sensitive to acidification.

A review of the information on the effect of acidification on aquatic organisms in Atlantic Canada testifies that the largest changes in phytoplankton occur within the pH range from 4.7 to 5.6, and slight changes are observed at pH from 5.5 to 6.5. pH tolerance varies greatly among invertebrates, but the average permissible pH for the majority of sensitive species lies within the range of 5.2 to 6.1. Sensitive fish species are subjected to the effect at pH making up 6.0–6.5, which corresponds to ANC = 50 [24–30].

A. Henriksen [25] proposed a differentiated approach to the determination of ANC$_{limit}$, when its value is a function of the volume of the atmospheric deposition of $SO_4^{2-}$. If the atmospheric intake of $SO_4^{2-}$ is low, then ANC$_{limit}$ may be close to 0, and the proportion of precipitation acidity increases. The critical value of ANC$_{limit}$ may already be as high as 70–100 $\mu$eqL$^{-}$ in areas of significant deposition sulfate caused by smelters, while ANClimit may not exceed 20 $\mu$eqL$^{-1}$ in remote regions such as the northern territories of Scandinavia. Analysis of the criteria for assessing the effect of water acidification on water communities in local contamination of the Arctic region allows us to recommend keeping ANC$_{limit}$ = 50 $\mu$eq L$^{-1}$ as the limiting level.

### 4.4. Critical Loads and Their Exceeded

The steady state water chemistry (SSWC) method was used to determine critical loads (CL) [29]. The method is based on determining the numerical value of the natural buffer capacity for specific lakes and is found as the chemical weathering of cations from the catchment to the values that provide ANC$_{limit}$. The relevance of this method persists because it allows for a comparative assessment of the permissible precipitation of acid-forming substances from the atmosphere on the catchments. To calculate the CL and critical load exceeds (Clex) for the tundra lakes of the Kola North and Norilsk region, we applied the calculations similar to the ones performed for the Scandinavian countries using present water chemistry.

$$CL = ([BC_o^*] - [ANC_{limit}]) \, Q - BC^*_{dep}, \tag{3}$$

$$CLex = CL - SO_4^*{}_{dep} - NO_{3dep} + BC^*_{dep}, \tag{4}$$

where $BC_o^*$ is the natural saturation of waters with the base cations of a specific water body in the natural state, Q is the run-off from the catchment and $SO_4^*{}_{dep}$ is the deposition of technogenic sulfates, $NO_{3dep}$ of nitrates and $SO^*{}_{dep}$ of base cations.

The CL calculation method had to be adapted for the lakes of WS taking into account that chlorine got into the waters, not only as part of marine aerosols, but also from geological sources of the catchment and from even a more significant anthropogenic source (contamination during the development of oil and gas fields). In this case, the Cl not compensated by Na in the sea salts proportion contributes to the water acidification [10]. Another important point in the adaptation of the method was the calculation of the amount of nitrates received from the combustion of associated gas during oil fields exploration, a separation thereof from the natural ones that are part of the organic matter. The natural background for the nitrate content in the lake waters of the WS lakes is significantly higher than in the waters of the Kola North lakes. It is caused by both natural and anthropogenic factors (associated gas combustion during oil production). The minimum values of the nitrate concentrations allow for a calculation of their dependence on the concentrations of organic acids as the natural nitrate concentrations increase simultaneously with the increase in anions in organic acids, which is associated with the decomposition of organic substances:

$$[NO_3]o = 0.118 \, [A^{n-}]_t, \tag{5}$$

where $A^{n-}$ is the content of organic anions, which is empirically calculated:

$$A^{n-} = DOC \, (4.7 - 6.87\exp(-0.332 \, DOC)), \tag{6}$$

In the case of excess income of chlorine and nitrates, the calculation of the natural saturation of waters with base cations was as follows:

$$BCo = [BC]t - F \, (([SO_4]t[SO_4]o) + ([NO_3]t - [NO_3]o) + ([Cl]t - [Cl_{Na}])), \tag{7}$$

where $[Cl_{Na}]$ is the chlorine content in the water, compensated by the sodium content.

The dependence of the natural sulfate content on the base cation content is approximated by the following equation for lake waters in the tundra zone of WS:

$$[SO_4]o = 2.67 + 0.021 \, [BC]t, \; r = 0.72, \; p < 0.001; \tag{8}$$

Thus, the necessary conditions for estimating base cations entering from catchments providing neutralization of anthropogenic acids were determined. An amount of 50 $\mu eqL^{-1}$ was taken as a critical value ($ANC_{limit}$).

According to the data obtained, the CL values of about 5% of the lakes in the tundra of ER and forest-tundra do not exceed 0, and that of 75% of the lakes do not exceed 50 $\mu eqL^{-1}$ (Figure 4). No negative values of CL were found in WS tundra, while there are a little more than 20% of cases with critical values, which is explained by a lower load of acid-forming substances in these regions. No acidified lakes have been identified in ES. The critical load exceeds were 56% and 12.5% in the Kola region and in the north of Western Siberia, respectively (Figure 4).

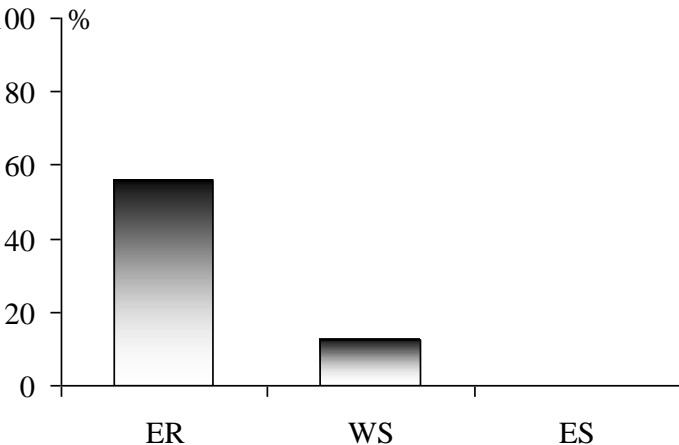

**Figure 4.** The percentage of lakes with critical loads exceeded in the Arctic regions of the European territory of Russia (ER), Western (WS) and Eastern Siberia (ES).

### 4.5. Metal Enrichment of Waters

Trace metals are components of natural waters, and their concentrations in waters vary depending on the geological conditions of the watershed. Regional deposition and transboundary pollution from the contaminated atmosphere increase the metal concentration in water [30–38]. Trace metals pose a potential hazard to vulnerable ecosystems in the Arctic. The international scientific community is mostly interested in studying the contents of such dangerous elements as Hg, Cd and Pb [10,11].

The variety of factors that affect the migration of elements complicates the assessment the contribution of anthropogenic factors to the enrichment of waters with trace elements, including toxic ones [30–38]. The analysis of the chemical composition of the waters of Arctic lakes has shown that airborne pollution by elements in the studied regions around industrial centers does not lead to severe water contamination but contributes to an increase in the concentration of many elements. At the same time, focusing on the water quality standard in the Russian "Maximum permeation concentration" data that are used in Russia from the Arctic zones to the steppe regions is not correct, taking into account the high vulnerability of the Arctic aquatic ecosystems.

Enrichment factors (EFs) are used as an effective method to assess the anthropogenic contribution; EF is used to assess the contamination of soils, surface layers of bottom sediments in lakes or the aquatic environment [28–30].

Using this approach, we calculated the EF for the waters of the studied lakes of the Arctic zone. It is worthy to note an important point—we calculate the percentage of certain elements in the mineral composition of the waters of the total sum of salts contained in the waters. Al was adopted as the reference element. We used the ratio for those rocks that are characteristic for a particular region: for the Kola North, we used data on the elemental composition of rocks according to A.P. Vinogradov [37] and for WS and ES of sedimentary rocks according to K.K. Turekian and K.H. Wedepohl [38]. The EF was calculated by the formula:

$$EF_{Cx} = (C_x:Al)_{water}/(C_x:Al)_{rock},\tag{9}$$

An analysis of the results has shown that higher values of EF for Ni, Cd, As, Sb and especially Se occur in the ER tundra zones, near non-ferrous metal smelting plants (Figure 5). The content of other rare elements is also increased, which may be caused by both their dispersal with smoke emissions and leaching by acid precipitation.

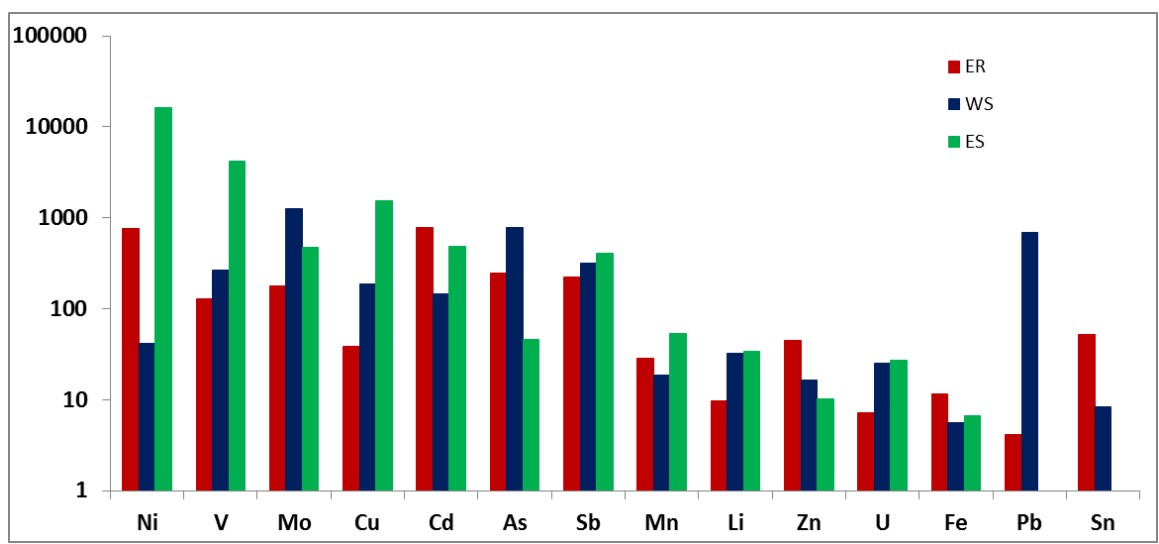

**Figure 5.** Enrichment factors of elements.

The EF is significantly lower for many elements in the WS waters. The greatest enrichment factors are determined for Pb, V, As, Sb and especially Mo. It is worthy to note that oil deposits are typical for WS, and as a result, oil is constantly leaking to the surface. Oil is known to contain a wide range of elements, especially V, As, Sb, Mo and others. Oil is biodegradable in the environment, while metals become included in the biogeochemical cycle, which inevitably affects the geochemistry of surface waters. Associated gas is also combusted here, entailing the dispersion of trace elements [11–20].

The content of some elements was below the detection limit (Sn, Se) in the lakes near Norilsk; therefore, they are absent in the diagram. The highest EF values were obtained for the Norilsk region for metals Re, Ni, V, Cu, Cd, Mo and Sb. The variability in the lead content in lakes near Norilsk is high—according to the calculation results, the maximum enrichment factor exceeds 1500. The enrichment factor for most elements in the Norilsk region exceeds that for ER and WS. It should be noted that the lakes studied were quite close to the Norilsk plant, respectively, and were more significantly affected by smoke emissions.

## 5. Conclusions

The chemical composition of the water in the lakes of the studied territories of the Arctic regions has both general and specific features, which are explained by the peculiarities of the geochemical structure of the catchments and the differences in anthropogenic impacts. All lakes are low mineralized and oligotrophic with a low content of nutrients. The highest salinity characterizes the waters of the Norilsk region, as a result of the proximity of the investigated lakes to the industrial zone and of the peculiarities of the geological structure.

Acidified lakes with low pH values, high transparency of waters and high sulfate levels are found in the north of ER and WS. Dystrophic lakes with high color of waters are typical for swamped catchments. The tundra watershed of the Kola region is the least resistant to acid depositions. Almost 75% of the studied catchments have high vulnerability and 5% of lakes have negative values of ANC. The ANC values are higher in the tundra of the north of WS.

Analysis of critical load excess showed that the percentage of lakes that have acidification potential is the highest (~ 60%) in the ER tundra and forest-tundra. The acidification potential is much lower—12.5% in WS; the lake waters in the Norilsk region around the industrial complex have a high buffering capacity. However, it is worthy to note that a small number of lakes around the plant were examined. Here, dust emission can also increase the buffer properties of waters.

The influence of the Nornickel concern production was manifested in increased concentrations of Ni and Cu in the lakes of the Norilsk and Kola regions. The contents of Al, Cd, Cs and Se are the

highest in the soft and ultra-soft waters of the ER Arctic region. It is known that the acid precipitation on the watershed recorded here leads to the intensification of elements leaching from the underlying rocks. The highest concentrations of Pb were found in the waters of the lakes around the Norilsk plant, which reflects the influence of the industrial and urban complex. The waters of the lakes of Western Siberia are also characterized by higher contents of Cu, V, Sb and As. The calculation of the enrichment factor allowed us to conclude that the highest EF values were obtained for the Norilsk region (ES) for Re, Ni, V, Cu, Cd, Mo and Sb. For the WS waters in the tundra and forest-tundra zones, the highest enrichment factors are for Pb, V, As, Sb and especially Mo. In the lakes of the ER tundra and forest-tundra zones, Ni, Cd, As, Sb and especially Se have higher EF values, which reflects the influence of industrial activity in this region.

The study of small lakes as indicators of airborne contamination of the watershed has shown that, caused by local emissions and transboundary pollution transfers to the Arctic regions, acidification processes occur in acid-sensitive areas (in terms of geological conditions of catchment) and increase the content of metals, including toxic ones. Although the concentrations of these elements do not exceed the highest permissible concentrations for drinking water, it is a signal of adverse changes. Local studies in the three Arctic regions of Russia showed the need for more detailed territorial studies of the land waters of the tundra regions.

**Author Contributions:** Conceptualization, T.I.M.; investigation, N.A.G. and M.I.D.; methodology, N.A.G., M.I.D. and T.A.K.; project administration, T.I.M.; software, M.I.D.; visualization, V.Y.K. All authors have read and agreed to the published version of the manuscript.

**Funding:** The work was supported by grants from the Russian Fund for Fundamental Investigation (Grant 18-05-60012/18). The study was supported by the Ministry of Education and Science of the Russian Federation (Federal Research Program 0137-2019-0008).

**Conflicts of Interest:** The authors declare no conflict of interest.

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
