# Peer review of "Water Chemistry of Arctic Lakes under Airborne Contamination of Watersheds"

_water, doi:10.3390/w12061659_

Round 1

Reviewer 1 Report

This manuscript introduced the water chemistry of arctic lakes under airborne contamination of watersheds. The topic of this study is interesting. But some parts still need to be improved. Here are some specific comments.
(1) The literature review needs to be more critical.
(2) How to determine the source is from airborne contamination? How about the contaminants from surface input?
(3) What is the different feature for this study area?
(4) How about the impact of weather/climate conditions?
(5) More details about statistical analysis in result analysis is required
(6) Please discuss more about the mechanism in this process.
(7) Please compare the results in this study with those in previous studies.
(8) Which factor would play the most important roles for the obtained results?

Author Response

The authors are grateful for the positive assessment of the manuscript and valuable comments.

Reviewer 2 Report

The article is good scientific material which shows that increasing sources of pollution, their quantity and quality affect the quality of water, in this case lakes, and attention should be paid to this.

The authors have analyzed water samples in terms of many water quality parameters from the three regions, which showed that the chemical composition of water is influenced by the geology of the ground, the location and the use of industrial plants.

The essence of the study is that acidification of these waters is not only related to the existing industry, but this effect is influenced by other factors. The activities aimed at the lack of protection of the plants against the emission of pollutants, such as purification zones in the areas directly adjacent to them, should not be underestimated.

The results obtained and the analysis in the paper presented confirm that it is necessary to act systematically in the environment.

I have only a few remarks to figure 1 to enlarge what is important, to introduce the legend for clarity. I ask the authors to change the units recorded at chemical values in the already existing SI system.

I have a request for the authors to supplement the latest literature from 2019 with a discussion of research results. This will certainly enrich the work. 

Author Response

(The authors gave the same response as above.)

Round 2

Reviewer 1 Report

The authors have improved the manuscript.